# Deep learning-based diagnosis of feline hypertrophic cardiomyopathy

Jinhyung Rho[1,2], Sung-Min Shin[3], Kyoungsun Jhang[4], Gwanghee Lee[4], Keun-Ho Song[5], Hyunguk Shin[6], Kiwon Na[7], Hyo-Jung Kwon[5], Hwa-Young Son[5]*

1 Jeonbuk Pathology Research Group, Korea Institute of Toxicology, Jeonbuk, Republic of Korea, 2 Center for Companion Animal New Drug Development, Korea Institute of Toxicology, Jeonbuk, Republic of Korea, 3 Do Animal Hospital, Incheon, Republic of Korea, 4 Department of Computer Engineering, Chungnam National University, Daejeon, Republic of Korea, 5 College of Veterinary Medicine, Chungnam National University, Daejeon, Republic of Korea, 6 24 Africa Animal Medical Center, Daejeon, Republic of Korea, 7 Daejeon Central Animal Medical Center, Daejeon, Republic of Korea

* hyson@cnu.ac.kr

**Data Availability Statement:** All relevant data are available on Figshare: 10.6084/m9.figshare. 21128266.

**Funding:** This study was supported by the Chungnam National University (2020-1438-01).

## Abstract

Feline hypertrophic cardiomyopathy (HCM) is a common heart disease affecting 10–15% of all cats. Cats with HCM exhibit breathing difficulties, lethargy, and heart murmur; furthermore, feline HCM can also result in sudden death. Among various methods and indices, radiography and ultrasound are the gold standards in the diagnosis of feline HCM. However, only 75% accuracy has been achieved using radiography alone. Therefore, we trained five residual architectures (ResNet50V2, ResNet152, InceptionResNetV2, MobileNetV2, and Xception) using 231 ventrodorsal radiographic images of cats (143 HCM and 88 normal) and investigated the optimal architecture for diagnosing feline HCM through radiography. To ensure the generalizability of the data, the x-ray images were obtained from 5 independent institutions. In addition, 42 images were used in the test. The test data were divided into two; 22 radiographic images were used in prediction analysis and 20 radiographic images of cats were used in the evaluation of the peeking phenomenon and the voting strategy. As a result, all models showed > 90% accuracy; Resnet50V2: 95.45%; Resnet152: 95.45; InceptionResNetV2: 95.45%; MobileNetV2: 95.45% and Xception: 95.45. In addition, two voting strategies were applied to the five CNN models; softmax and majority voting. As a result, the softmax voting strategy achieved 95% accuracy in combined test data. Our findings demonstrate that an automated deep-learning system using a residual architecture can assist veterinary radiologists in screening HCM.

## Introduction

Feline hypertrophic cardiomyopathy (HCM) is a common, chronic, and life-threatening heart disorder that affects 14.7% of cats aged > 7 months [1]. Unlike humans, the most frequent, predominant cardiac disease is the HCM. In 306 primary cardiac disorders from 1998 to 2005, 252 cases were cardiomyopathy (82%) and 48 cases (16%) were congenital heart disease [2, 3]. It is also among the ten most common causes of death in cats [4]. Feline HCM causes heart enlargement, with prominent ventricular wall hypertrophy and interventricular septum.

The funders had no role in study design, data collection and analysis, decision to publish, or preparation of the manuscript.

**Competing interests:** The authors have declared that no competing interests exist.

Consequently, the left ventricular cavity is narrowed and the left atrium is dilated, in the heart of the affected cats [5]. Clinical signs of HCM include hyperventilation, syncope, and arterial thromboembolism [1]. Moreover, 25% of cats aged > 9 years are asymptomatic [4]. However, the most ominous sign of HCM is sudden death, which may occur within seconds and without any notable symptoms [6].

Various diagnostic tools for HCM include physical examination, electrocardiography, ultrasound examination, X-ray imaging, and blood analysis [7]. Initially, incidental findings on physical examination showing murmur, gallop sound, or arrhythmia can be considered. Additionally, evaluating the levels of N-terminal-pro brain-type natriuretic peptide (NT-proBNP) may be performed; however, radiology and echocardiography are considered the gold standard for diagnosing HCM. Specifically, a radiographic thoracic ventrodorsal (VD) image of a cat with feline HCM shows a specific cardiac silhouette known as the valentine heart. Several methods for radiographical measurement of heart enlargement are prevalent, including vertebral heart size, modified vertebral heart size, and cardiothoracic ratio. Although quantitative, VHS shows only 51% accuracy in feline radiography, indicating that many examinations should be combined to confirm the HCM [8].

Machine learning (ML) is a dominant paradigm in radiologic and histological analyses. By combining computer science and statistics, ML allows self-learning and improves performance by learning from experience. Deep learning (DL) is an ML class involving learning through neural networks with multiple representation levels corresponding to different abstraction levels [9]. Owing to its high accuracy, there are various approaches regarding the application of DL in veterinary diagnostics, including cytology [10], fecal parasite detection [11], histopathology [12], and radiology [13]. Tommaso et al. attempted to classify canine radiographic findings on thoracic lateral (LL) images and observed a sensitivity > 90% [11]. However, few studies have investigated DL-based feline radiographic findings analyses. Here, we investigated the DL-based classification of feline HCM in 275 thoracic VD radiographic images from Chungnam National University Veterinary Hospital and local veterinary hospitals using five deep neural networks, with validation through diagnosis by a radiology specialist, and to determine the most optimal deep neural network model.

## Materials and methods

### Dataset

Image quality, proper positioning, and exposure were considered by a veterinary radiology specialist, and only fine images were selected. For the generalizability of the dataset, we obtained 273 feline thoracic VD X-ray images obtained from independent 5 institutions; Chungnam National University Veterinary Medical Teaching Hospital and four local animal hospitals. The institutions providing radiographic images are enlisted in Table 1. For personal reasons, the name of the institution is sealed. All radiographic images were obtained during a routine examination or follow-up. Owner consent was obtained to use these images for the study. Only images and disease states (normal or HCM) were provided to the researcher. All images were anonymized and diagnosed by a veterinary radiology specialist and approved for

**Table 1. The information of the dataset used in the experiment.**

| Animal Hospital / Case | A | B | C | D | E (revision) | Total |
|---|---|---|---|---|---|---|
| **Normal** | 32 | 30 | 27 | 10 | 10 | 109 |
| **HCM** | 55 | 50 | 34 | 15 | 10 | 164 |

research use. Among the 273 radiographic images, 164 and 109 were diagnosed as HCM and normal, respectively. 143 and 88 HCM and normal images were used for the learning process. Further, 21 images determined as HCM and 21 images of normal were sorted into test images for evaluation of models. In order to avoid the peeking phenomenon, we used 11 normal and 11 HCM images to analyze the accuracy of the model. Then, 20 images were additionally added to evaluate the peeking phenomenon and the voting strategy. S1 Table presents detailed information.

## Image mask production

Various open-source libraries were used for image processing and the learning process. A library operating system (OS) was imported to process the directory and files. Images were processed into arrays using NumPy and Matplotlib. TensorFlow, Keras, OpenCV, Pillow, and Scikit images were imported to deal with the deep neural network model and model. After learning, we used an open-source Pyplot to plot the graphs and confusion metrics. Fig 1 shows the detailed process of the image analysis. The image mask comprised one channel constituting 0 and 1 per pixel. Specifically, the heart was indicated by 1 while the other pixels were expressed as 0, which allowed the computer to recognize the heart's location. To maximize the accuracy, Unet was implemented for image mask production. Unet is a network architecture used for precise image segmentation. First, X-ray images were resized to $256 \times 256$ and converted into arrays with three numbers expressing red, green, and blue intensities in each pixel (three channels). The shape of the image array was expressed as height, width, and channels, while the status of the converted image was expressed as (256, 256, 3). To train the Unet, previously marked masks were prepared and processed together. Consequently, an image mask was produced, indicating the heart location (256, 256, 1). Mask images were further processed using a Binary threshold and Otsu threshold to enhance accuracy.

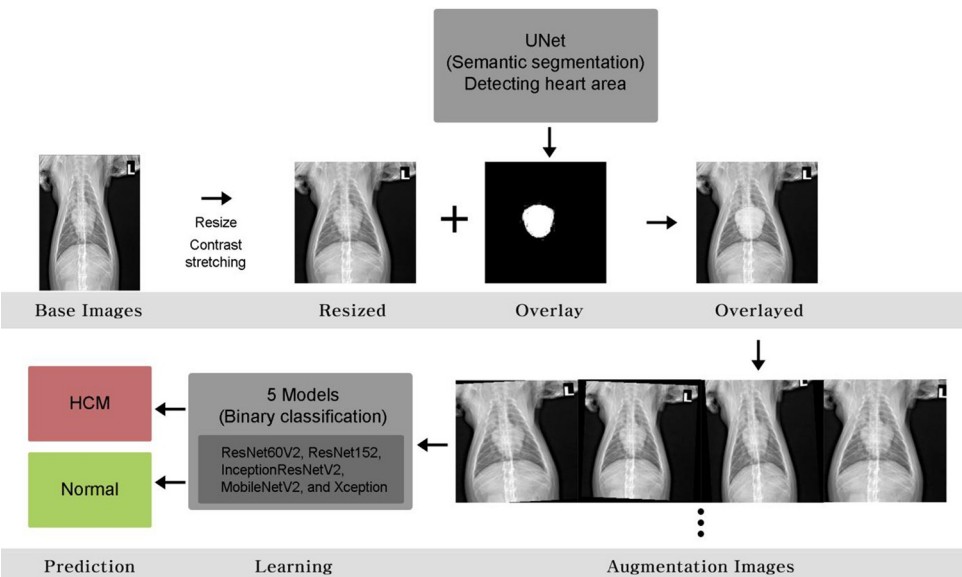

**Fig 1. Schematic illustration of the learning process.**

## Pre-processing procedures

X-ray images for training were resized to 233 × 233 and converted to arrays (233, 233, 3), followed by pre-processing to enhance accuracy. Regarding pre-processing, we implemented Contrast-stretch and Overlay using a previously produced image mask. Contrast stretching extends a specific contrast range to enhance image distinction. Overlay with an image mask indicating the heart area improved heart detection and diagnosis. During the overlay process, the image mask was resized to 233 × 233 to fit the original images.

## Data augmentation

The pre-processed images were amplified by data augmentation, which is a valuable tool to increase the diversity of a training set by applying random transformations. It is useful when there is insufficient data or a low detection rate. In our dataset, the original training dataset images were randomly rotated in 5-degree intervals, sheared in the range of 0.2, and zoomed in the range of 0.2, respectively, to obtain the final training dataset images, which are 2958 images of normal and 2860 images of HCM.

## Image classification

After pre-processing, five deep neural network models (Resnet50V2, Resnet152, InceptionResnetV2, MobilenetV2, and Xception) were compared to investigate the optimal engine for HCM diagnosis from feline VD X-ray images. Application of the deep neural network model was performed using Opensource library TensorFlow and Keras. The learning rate was set to start from 0.000001 and increase linearly to 0.001 until the 13 epoch. From the 14th epoch, the learning rate decreased by an exponential decay. The repetition of learning was set to 45 times, i.e., epochs, and designed to stop whenever there was an improvement in the loss function, which prevented the overfitting of the learning. The learning process of five architectures began from imagenet pretrained weight. In each pre-trained model without a classifier, we built additional classifier layers constituting a flattening layer and four fully-connected dense layers. Between the dense layers, we used a dropout probability of 0.5 to prevent overfitting. After the learning process, we analyzed the accuracy and loss graphs; additionally, the model was tested using 22 X-ray images (11 normal and 11 HCM) not used in the learning process. Based on the test results, we drew a confusion matrix and receiver operating characteristic (ROC) curve, followed by a comparison of the accuracy for each deep neural network model. The weight h5 file and Colab files are included in supplementary files.

## Metrics to compare neural network architectures

Based on the learning process of five neural networks, we analyzed various factors including accuracy, precision, recall, F1 score, sensitivity, specificity, and area under the curve (AUC) score. The definitions for the comparison metrics are as follows:

Accuracy = Number of correct predictions/total number of predictions.

Precision = True positive/(true positive + false positive)

Recall (sensitivity) = True positive/(true positive + false negative)

Specificity = True negative/(true negative + false positive)

The F1 score represents the harmonic mean of precision and recall. The AUC score refers to the two-dimensional area underneath the entire receiver operating characteristic (ROC) curve.

### Ensemble strategy

Based on the request for the revision, we applied an ensemble strategy to enhance the model accuracy, prevent peeking, and minimize the misdiagnosis arising from a single architecture. We employed two strategies in the ensemble: majority voting and softmax voting [14]. Each model prediction, softmax output, is converted to 0 (HCM) or 1 (NORMAL) using the Argmax function. Majority voting is performed based on the sum of the odd number of binary prediction outputs. If the number of outputs is 5 and the sum is greater than or equal to 3, the diagnosis is NORMAL; otherwise, the diagnosis is HCM. In contrast, softmax voting adds any number of softmax outputs that are the two weight values for NORMAL and HCM. Then, the Argmax function is applied to the sum of softmax outputs to decide which is more significant between NORMAL weight sum and HCM weight sum. The diagnosis is performed using a larger value of summed weights in the softmax strategy.

### t-distributed stochastic neighbor embedding (t-SNE) visualization

The CNN model has high-dimensional feature data that cannot be visualized and plotted in a three-dimensional world. To visualize how the model distinguishes radiographic image data, t-distributed stochastic neighbor embedding was applied to the test datasets whose corresponding feature data in the model are converted and visualized into 2-d feature points. The TSNE in the Scikit Learn library was implied to the test data and plotted in Fig 7. The feature data used in t-SNE is gained before passing the flattening layer of each model.

### Hardware and software

For modeling and coding, Anaconda3 and Jupyter Notebook were operated on a workstation with an AMD Ryzen 9 3900x 12core processor, 64 GB RAM, and Geforce RTX 3080.

## Results

### Learning process analysis

Fig 2 presents the results of the learning process. Epochs refer to cycles of repeated learning. The number of cycles positively and negatively correlated with the training accuracy and loss rate, respectively. There was a gradual increase in the accuracy of all training models, as evident from the accuracy and loss graphs. However, regarding the validation accuracy of the ResNet50V2 model, an irregular peak in the 10th epoch was observed. Further, the validation accuracy of the ResNet152 showed three and two irregular peaks in the accuracy and loss graphs, respectively. The validation accuracy of the inceptionResNetV2 model showed two and three irregular peaks in the accuracy and loss graphs, respectively. Moreover, MobilenetV2 demonstrated two peaks in the accuracy and loss graphs, while Xception rarely showed irregular peaks in the graphs.

### Model test and evaluation

After the learning process, each model predicted the diagnosis of the previously sorted images. Figs 3 and 4 show the prediction results. The MobilenetV2 showed 90.91% of accuracy. The Resnet50V2, Resnet152, InceptionResNetV2, and Xception showed 95.45% accuracy respectively. Although the same accuracy, the misdiagnosed image of each model differed from each other (Figs 2 and 3) in 22 test images, except the Resnet50V2 and Resnet152 (Figs 2 and 3).

To analyze model performance, the ROC curve was analyzed (Fig 5), which shows the performance of the classification model at all classification thresholds. We calculated the accuracy,

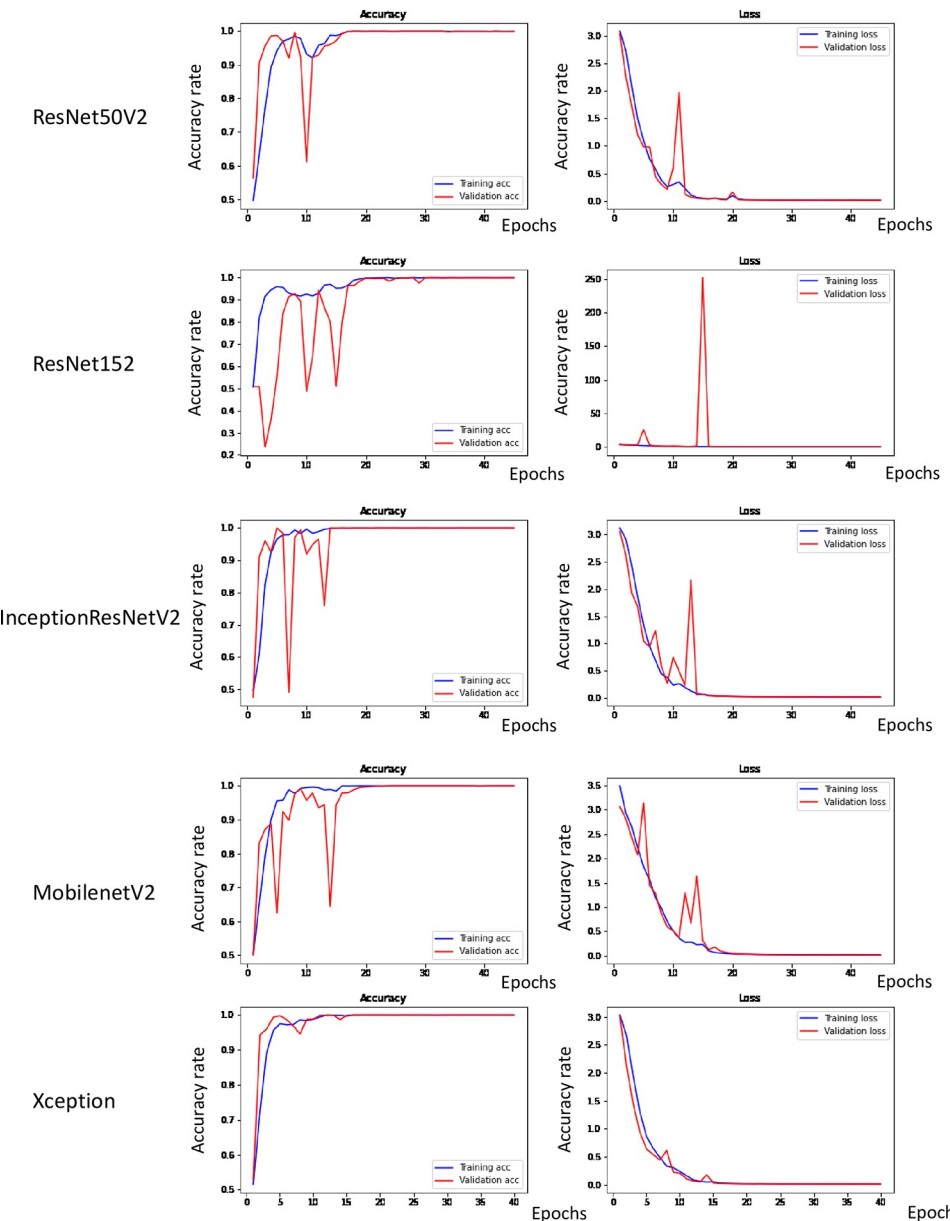

**Fig 2. Accuracy and loss graphs after training the architectures to detect feline HCM using VD X-ray images.** Five DL architectures were used: ResNet50V2, ResNet152, InceptionResNetV2, MobileNetV2, and Xception. Epochs refer to the repetition of learning.

precision, recall, F1 score, and area under curve (AUC) score (Table 2). ResNet50V2 showed 95.45% accuracy, 100% precision, 91% recall, 95% F1 score, and 75.2% AUC score. ResNet152 showed 95.45% accuracy, 100% precision, 91% recall, 95% F1 score, and 80.57% AUC score. InceptionResNetV2 showed 95.45% accuracy, 100% precision, 91% recall, 95% F1 score, and 87.6% AUC score. MobileNetV2 showed 91.91% accuracy, 100% precision, 82% recall, 90% F1 score, and 87.6% AUC score. Finally, Xception showed 95.45% accuracy, 92% precision, 100% recall, 96% F1 score, and 91.63% AUC score.

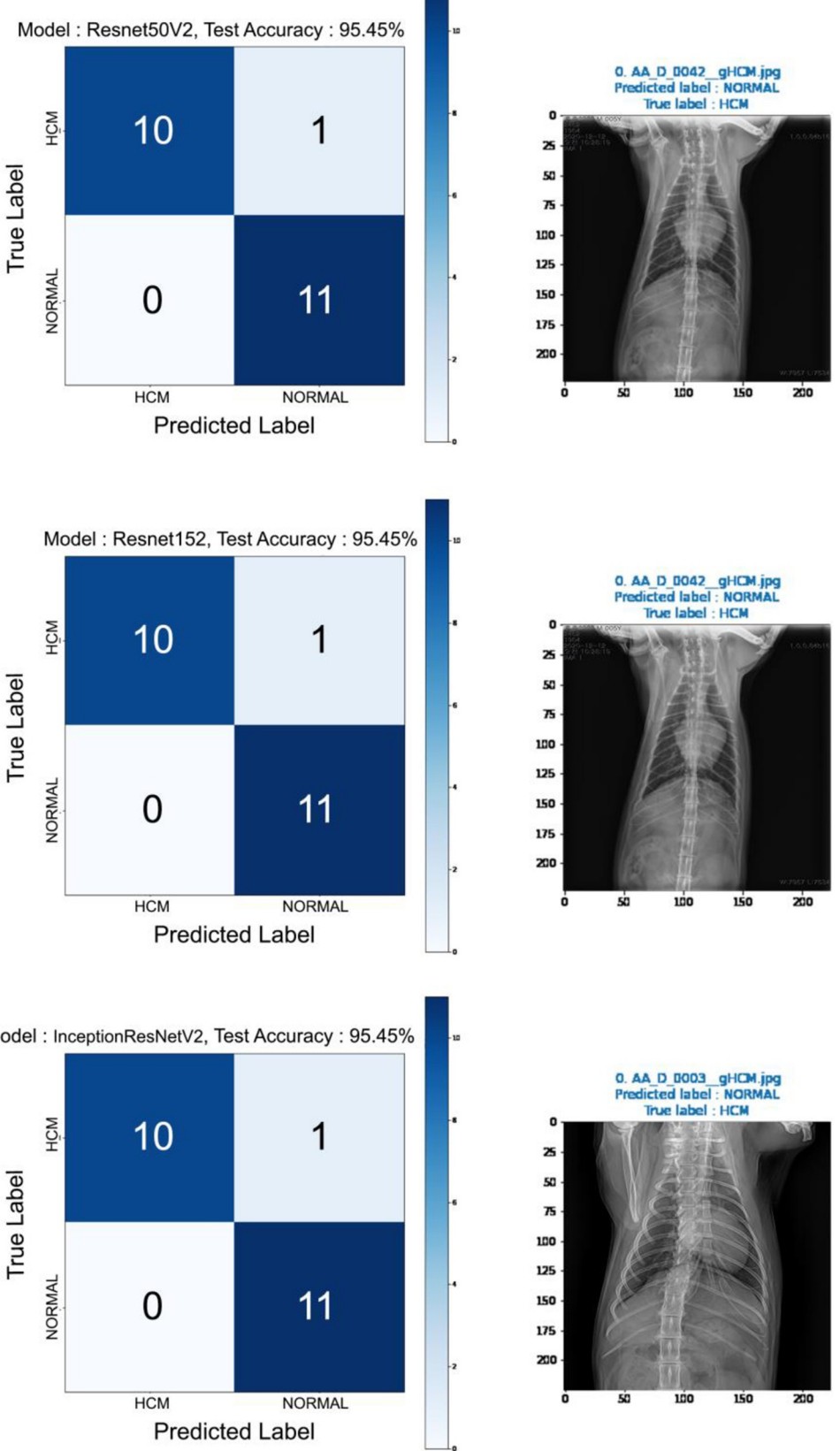

**Fig 4. Confusion matrix obtained from the deep learning process of detecting feline HCM using VD X-ray images (ResNet50V2, ResNet152, and InceptionResNetV2). Cont'd.** Twenty-two X-ray images were tested. X-ray images on the right indicate original images that the architecture could not classify.

## Testing on additional data

One of the biggest concerns in Deep learning is "peeking." Peeking is a severe problem in which the learning process influences the test data, consequently showing high accuracy in test data but erroneous outcomes in the unseen data. Peeking is inevitable, yet we tried to avoid it.

To evaluate peeking in the study, we obtained normal and HCM-affected radiograph images (10 each) from an uninvolved vet hospital. We evaluated the model from the new data and compared it to the previous data (Fig 6). The test result of accuracy in new data showed 75% in Resnet50V2, 70% in Resnet152, 85% in InceptionResNetV2, 55% in MobilenetV2, and 80% in Xception respectively. The combined results were 86% in Resnet50V2, 83% in Resnet152, 90% in InceptionResNetV2, 76% in MobilenetV2, and 86% in reception respectively.

To evaluate whether the model distinguishes the test dataset, we plotted the t-SNE of each model in Fig 7. As a result, the plotted test data in InceptionResnetv2, Resnet152, and Xception

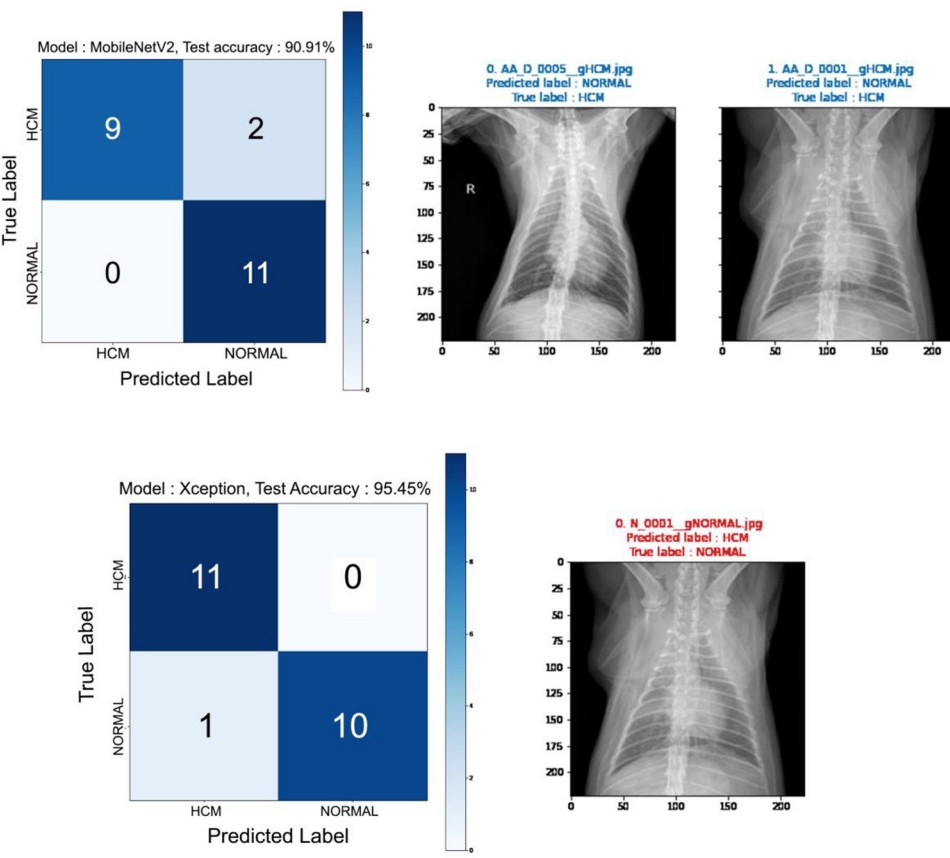

**Fig 4. Confusion matrix obtained from the deep learning process of detecting feline HCM using VD X-ray images (ResNet50V2, ResNet152, and InceptionResNetV2). Cont'd.** Twenty-two X-ray images were tested. X-ray images on the right indicate original images that the architecture could not classify.

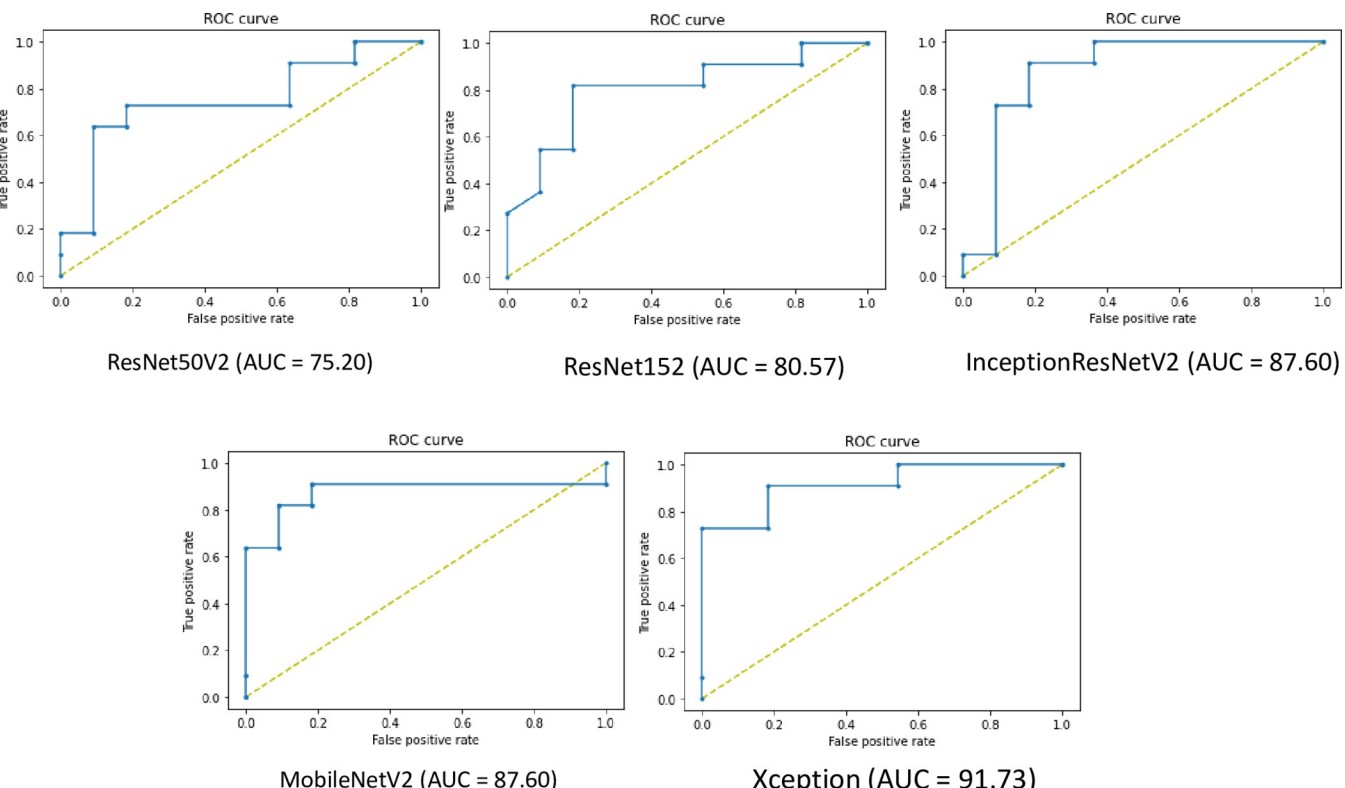

**Fig 5. ROC curve of DL results in detecting feline HCM using VD X-ray images.** Five DL architectures were used and compared (ResNet50V2, ResNet152, InceptionResNetV2, MobileNetV2, and Xception). The AUC was calculated and presented.

showed a distinct separation between normal and HCM. The accuracy and loss results are enlisted in S2 Table.

### Ensemble strategy

To enhance the accuracy, minimize the misdiagnosis of a single model, and prevent peeking, we ensembled the results from each model in two ways: Majority voting and Softmax. We first tested in all test sets, including old and new data (Fig 8A). Interestingly, majority voting and softmax voting achieved 100% accuracy in old data. In the new data, however, the accuracy of majority voting achieved only 85%, whereas softmax voting achieved 90%. In combined results, the accuracy of majority voting achieved 93% and softmax 95%, respectively. The confusion matrix of the softmax voting strategy on combined test data is plotted in Fig 8B.

**Table 2. Evaluating parameters of each deep neural network model (%)—accuracy, precision, recall, F1 score, AUC score, sensitivity, and specificity.**

| Architecture | Accuracy | Precision | Recall (sensitivity) | F1 Score | AUC score | Specificity |
|---|---|---|---|---|---|---|
| ResNet50V2 | 95.45 | 100 | 91 | 95 | 75.20 | 100 |
| ResNet152 | 95.45 | 100 | 91 | 95 | 80.57 | 100 |
| InceptionResNetV2 | 95.45 | 100 | 91 | 95 | 87.60 | 100 |
| MobileNetV2 | 90.91 | 100 | 82 | 90 | 87.60 | 100 |
| Xception | 95.45 | 92 | 100 | 96 | 91.73 | 90.91 |

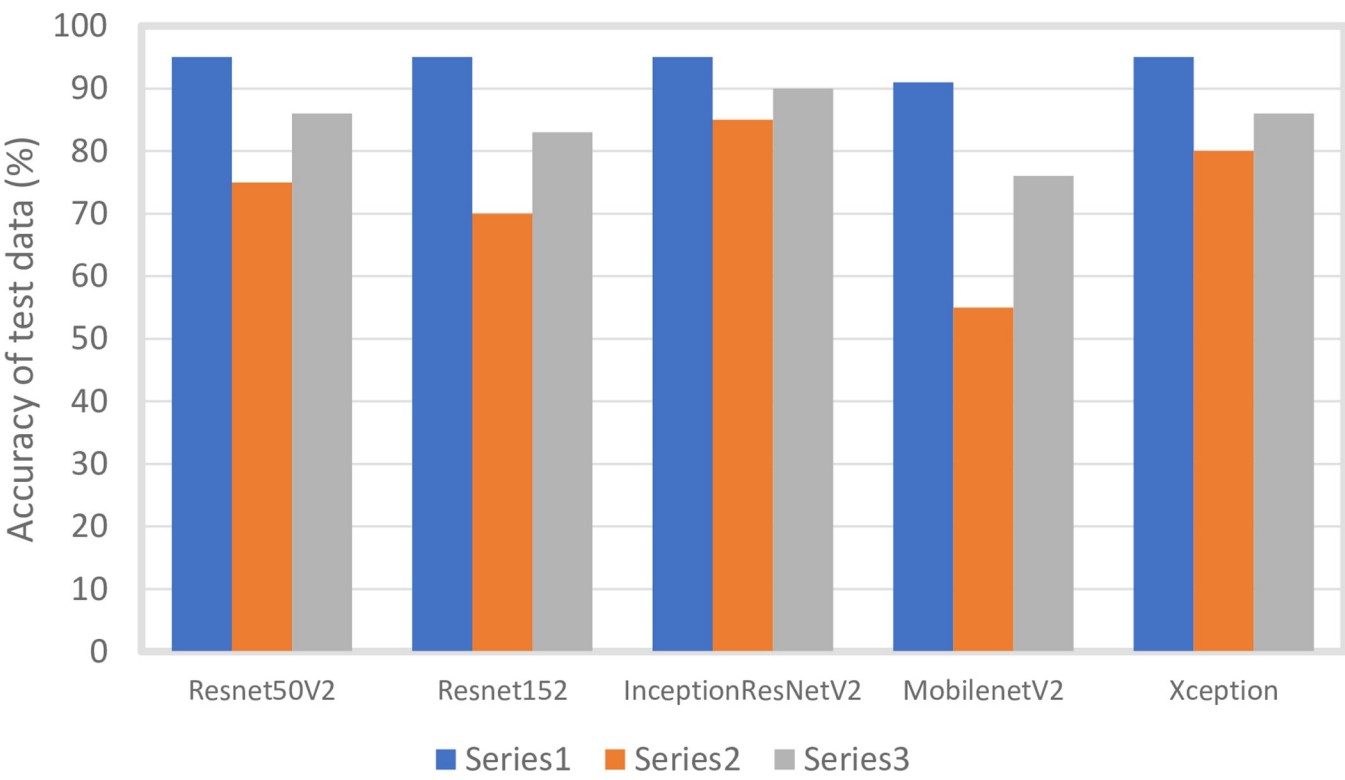

**Fig 6. Comparison of accuracy of five DL models in old test data, new data, and combined results.** Five DL architectures were used and compared (ResNet50V2, ResNet152, InceptionResNetV2, MobileNetV2, and Xception).

## Discussion

Recently, there has been a tremendous advancement in visual recognition in ML owing to the implementation of neuron architecture similar to the response of a neuron in the visual cortex, called a convolutional neural network (CNN) [15]. The Recent CNN architecture comprises the optimized setting of the convolutional structure. Although many architectures are available, we trained five residual learning frameworks (Resnet50V2, ResNet152, InceptionRes-NetV2, MobileNetV2, and Xception) using 231 images of feline VD X-ray images and determined the most optimal engine for diagnosing feline HCM. The accuracy of all engines was > 90%, with Xception being the ideal architecture.

ResNet152 (deep residual networks) comprises 152 depth layers but has lower complexity [16]. The ResNet, developed by Microsoft, is deeper than VGG nets, developed in 2014 by the Visual Geometry Group at Oxford University. Resnet152 has achieved excellent recognition tasks in ImageNet and Ms. COCO competitions. Specifically, ResNet152 has markedly lowered the top five errors encountered compared with the previous neural network. In our study, ResNet demonstrated > 90% accuracy; however, it showed several fluctuating patterns during the learning phase, as illustrated in the accuracy and loss graphs (Fig 2). ResNet50V2 is a 50-layered deep residual network developed by the inventors of ResNet 152. It has a two-layer block in a 34-layer net, with a three-layer bottleneck block.

InceptionResNetV2 was incepted in conjunction with ResNet [17]. It introduced residual connection with traditional architecture and demonstrated high performance in the 2015 ImageNet Large Scale Visual Recognition Challenge (ILSVRC). Inception is a convolutional network that was introduced by Google in the 2014 ILSVRC. It was subsequently modified into

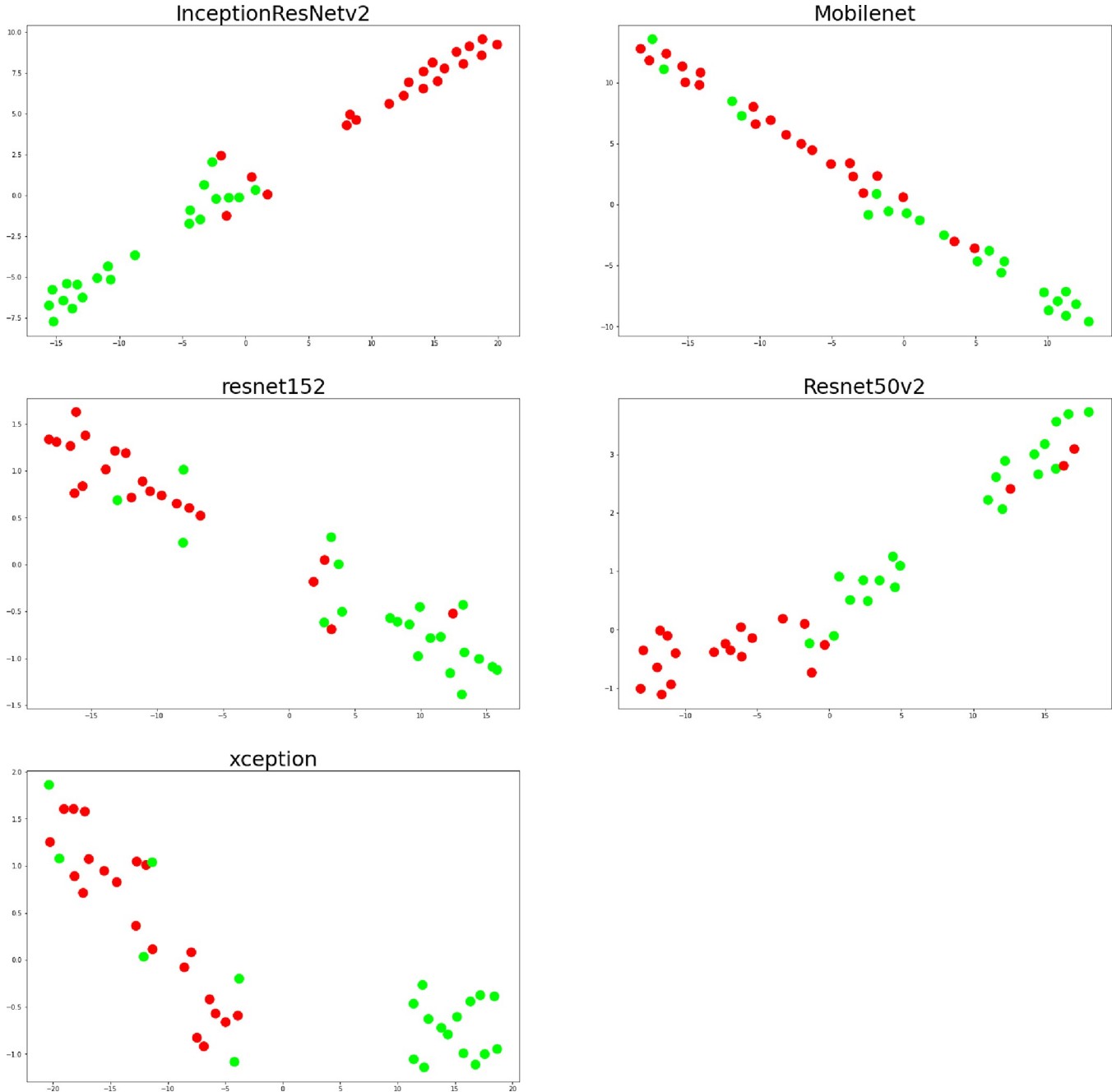

**Fig 7. t-SNE plotting of five deep learning models in all test data.**

version 4 (Inception V4). InceptionResNetV2 is a combination of InceptionV4 and ResNet. By conjugating both architectures and the correction process, there was a significant error reduction of 4.9%. In our study, InceptionResNetV2 showed 95.45% accuracy in the test; however, it fluctuated in the validation accuracy and validation loss graph.

MobileNet was designed in 2017 by Google for mobile environments; the architecture is focused on lightening the weight and enhancing efficiency [18]. To achieve efficiency, depthwise separable convolution was applied, which uses a single filter for each input channel, reducing

A

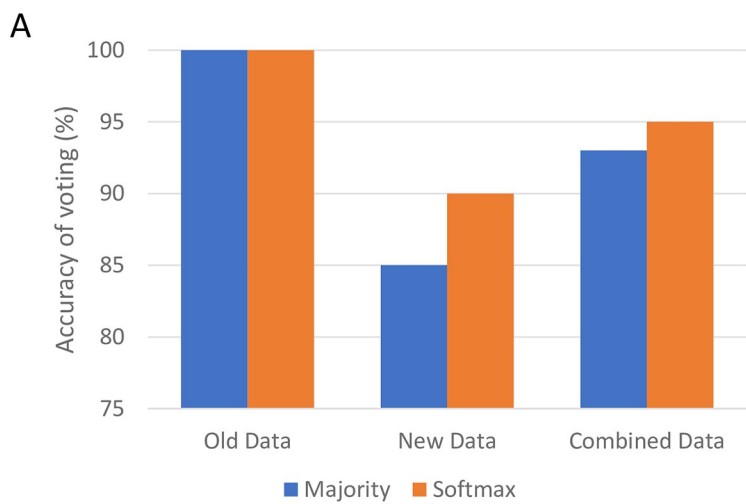

B

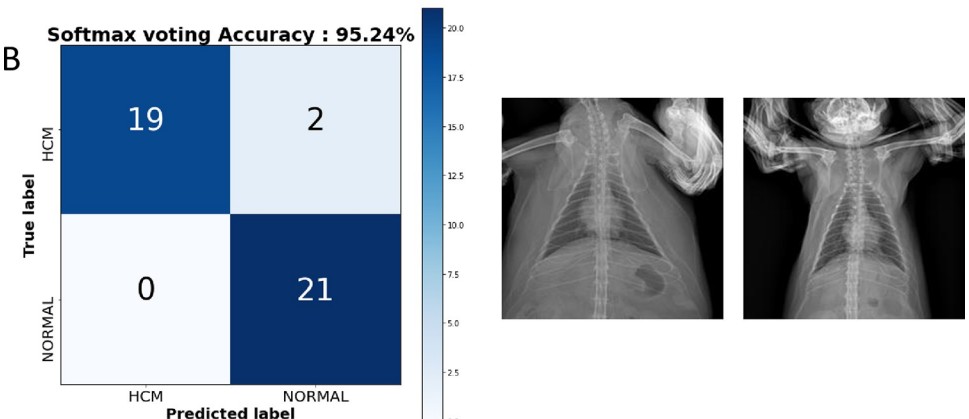

**Fig 8. Comparison of the accuracy of Majority voting strategy and Softmax strategy in old data, new data, and combined results** (A) and Confusion matrix of Softmax voting strategy and misdiagnosed images (B).

computation and model size. MobileNet uses linear bottlenecks, which have lower information loss and computation than ReLU. In our study, MobileNet showed an accuracy of 0.91.

Xception was introduced in 2017 by Google. Inspired by Inception, Xception applied Francois Chollet-modified depthwise separable convolution to independently calculate cross-channel correlations and spatial correlation, termed extreme inception [19]. Xception is composed of 14 modules and 36 convolutional layers. Specifically, the absence of a nonlinearity activation function increases the accuracy, including ReLU and ELU. In our study, Xception showed the smoothest accuracy, loss graphs, and the highest ROC curve. Based on the results, the most optimal architecture for diagnosing feline HCM was deduced to be Xception.

With the increase in computer capacity and processing ability, there has been extensive research on the application of DL in various veterinary diagnostic fields. Hattel et al. developed a two-staged algorithm that distinguished normal findings from several pathological diagnoses in the bovine liver, lung, spleen, and kidney stained using hematoxylin and eosin (H&E) [20].

The algorithm was based on a support vector machine (SVM), among the algorithms used in Ml through classification and regression analysis. Several rat studies have focused on evaluating H&E-stained bone marrow. Kozlowski et al. developed an automated algorithm for quantifying bone marrow cell density [12]. They suggested that automated measures of bone marrow cellular depletion were more accurate than manual scoring by pathologists. This algorithm was extended to quantify the myeloid, erythroid, lymphoid, and megakaryocyte cell density. With the improvement in digital pathology, Kartasalo et al. investigated the three-dimensional reconstruction of H&E-stained slides [21]. Specifically, they reconstructed the prostate and liver by annotating 2448 landmarks in the slides. In addition to histology, the Vetscan algorithm detects feline fecal parasites, including *Toxocara*, *Trichuris*, *Ancylostoma*, and taeniid eggs [11]. Additionally, hemosiderophages in equine bronchoalveolar lavage fluid were quantified based on a DL algorithm [10]. Regarding veterinary radiography, Banzato et al. developed an algorithm using DenseNet 121 and ResNet50 to detect radiographic findings in canine lateral (LL) X-ray images [13]. The findings were classified as cardiomegaly, alveolar, bronchial, interstitial, mass, pleural effusion, pneumothorax, and megaesophagus. Although some findings showed an accuracy of < 70%, some datasets detected cardiomegaly with a 98% accuracy.

Feline HCM is a multifactorial disease with various diagnostic protocols, including physical examination, genetic testing, NT-proBNP and troponin-I levels, blood pressure measurement, radiography, and ultrasound examination [7]. Recently, numerous biomarkers have been explored to diagnose feline HCM, including AIA, APOM, CPN1, prothrombin, etc. [22]. The gold standard for diagnosing HCM in cats is radiography and ultrasound analysis. Radiography of cats can identify HCM through symptoms, including cardiomegaly, auricular bulge, and silhouette changes [7]. Thoracic radiography allows the identification of mild or moderate cardiac changes in cardiomyopathy. For objective diagnosis of cardiomegaly in dogs and cats, radiologists implement a vertebral heart scale (VHS) [23]. By calculating the relative ratio of the long and short axis of the heart to vertebral size, a radiologist can determine whether the heart is enlarged. In addition, a modified VHS method enables the measurement of left atrial size in the lateral view of thoracic radiography. However, the combination of these indexes yielded an accuracy of only 75% [8]; therefore, various diagnostic protocols are recommended for confirming feline HCM. In our study, the diagnostic accuracy of thoracic radiography for feline HCM was 95%. Although sole dependence on thoracic radiography to diagnose feline HCM is risky, the model can suggest the possibility of feline HCM with at least 95% accuracy.

Another importance of the detection of HCM in a radiographic image is that it could save cost and increase the detection rate during the routine examination. As the only specialized veterinarian can perform the echocardiogram examination and high-cost equipment is required, an x-ray and auscultation are the most frequent tool to detect the HCM of the cat in the local animal hospital. By advising the vet whether the cat could be affected HCM by radiographic images, local vets can consider whether the animal should be transferred to the specialist.

This study has some limitations including the limited number of samples and the error rate. Because the subject of the study is specified into feline cardiomyopathy and the ventrodorsal (VD) radiographic images, comparatively low number of the datasets were available. However, we overcame the limited number of datasets by implementing the data augmentation and the voting strategy. Further investigation is required to obtain the more dataset to update the study and enhance the accuracy. Although the voting strategy of 5 models achieved 95% in the test data, the models misdiagnosed 2 images of HCM data as normal (Fig 8). To avoid those errors, the diagnosis should not only rely on a single examination.

A good CNN model for diagnosing HCM must distinguish HCM from normal heart distinctively. To visualize how the model deals with test data, we plotted the two-dimensional graph of

how the model recognizes test data. As the model is composed of high dimensions that cannot be plotted, the t-SNE function in the scikit learn library was implemented [24]. The accuracy of Mobilenet was only 76% because the model did not separate HCM from normal data. Further research is required to improve the separation of the HCM data from normal.

Peeking is one of the most challenging problems encountered by all DL scientists. Peeking is associated with overfitting in which the repeated modification of model weights based on test data causes grave outcomes in new data. Many computer scientists tried several attempts to overcome the peeking phenomenon [25, 26]. In this study, we tried to prevent the peeking phenomenon by 1) dividing the training dataset, validation dataset, and training dataset, 2) using the dropout function on each deep layer, and 3) evaluating new data that had never been used in training, validating and testing process. Despite all the precautions, we experienced the peeking phenomenon in every model. To improve the accuracy and reduce misdiagnosis of a single model, we tried the voting strategy as advised by the reviewer. As a result, we improved the accuracy by 95% in all test data combined.

## Conclusion

Our findings demonstrated that the five DL architectures provided 95% accuracy in all test data. Due to the peeking phenomenon, all models showed lower accuracy in the new test data and Mobilenet even showed 55% of accuracy in the new data. However, by implementing a softmax voting strategy using all 5 models, 95% accuracy in combined test data has been achieved. In conclusion, The automated DL system achieved high performance and could help local veterinarians screen HCM radiographically.

## Supporting information

**S1 Table. The category of the dataset used in the experiment.**
(DOCX)

**S2 Table. The accuracy, of 5 models on additional test data and combined results.**
(DOCX)

## Acknowledgments

This study is supported by the Chungnam National University and the Korea Institute of Toxicology (KIT).

## Author Contributions

**Conceptualization:** Hyo-Jung Kwon, Hwa-Young Son.

**Data curation:** Keun-Ho Song, Hyunguk Shin, Kiwon Na.

**Methodology:** Hwa-Young Son.

**Software:** Sung-Min Shin, Kyoungsun Jhang, Gwanghee Lee.

**Supervision:** Hyo-Jung Kwon.

**Validation:** Sung-Min Shin.

**Visualization:** Kyoungsun Jhang, Gwanghee Lee.

**Writing – original draft:** Jinhyung Rho.

**Writing – review & editing:** Jinhyung Rho.

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
