## [Decision Letter · Decision Letter 0]

4 Jul 2022

PONE-D-22-14859Deep learning-based diagnosis of feline hypertrophic cardiomyopathy: Comparison of five neural enginesPLOS ONE

Dear Dr. Young Son,

Thank you for submitting your manuscript to PLOS ONE. After careful consideration, we feel that it has merit but does not fully meet PLOS ONE’s publication criteria as it currently stands. Therefore, we invite you to submit a revised version of the manuscript that addresses the points raised during the review process.

ACADEMIC EDITOR: Concentrate on the dataset, it’s size and the ground truth more to validate your resultsPlease submit your revised manuscript by Aug 18 2022 11:59PM. If you will need more time than this to complete your revisions, please reply to this message or contact the journal office at plosone@plos.org. Please include the following items when submitting your revised manuscript:A rebuttal letter that responds to each point raised by the academic editor and reviewer(s). You should upload this letter as a separate file labeled 'Response to Reviewers'.A marked-up copy of your manuscript that highlights changes made to the original version. You should upload this as a separate file labeled 'Revised Manuscript with Track Changes'.An unmarked version of your revised paper without tracked changes. You should upload this as a separate file labeled 'Manuscript'.

We look forward to receiving your revised manuscript.

Kind regards,

Jude Hemanth

Academic Editor

PLOS ONE

Journal Requirements:

"NO authors have competing interests"

Reviewers' comments:

Reviewer's Responses to Questions

**Comments to the Author**

1. Is the manuscript technically sound, and do the data support the conclusions?

Reviewer #1: Yes

Reviewer #2: Yes

2. Has the statistical analysis been performed appropriately and rigorously? 

Reviewer #1: Yes

Reviewer #2: Yes

3. Have the authors made all data underlying the findings in their manuscript fully available?

Reviewer #1: Yes

Reviewer #2: Yes

4. Is the manuscript presented in an intelligible fashion and written in standard English?

Reviewer #1: Yes

Reviewer #2: Yes

5. Review Comments to the Author

Reviewer #1: Summary: Rho et al attempt to use deep learning networks to diagnose hypertrophic cardiomyopathy from radiographs of cats. Networks were trained on 231 radiographs (143 HCM vs 88 normal) and tested on a hold-out dataset of 22 radiographs (11 HCM vs 11 normals). Five different off-the-shelf network types were compared and all five gave identical results: 95% diagnostic accuracy (compared to 75% without AI).

Review:

I disclose that I have no expertise in vetenary/feline medicine but there does seem to be significant utility in such an algorithm. The manuscript is easy to follow and understand.

There are however some limitations:

1. the numbers of both training and test data (only 11 diseased subjects) is small. Probably too small to judge the algorithm's effectiveness on.

2. the validation set seems to be from the same institution and not independent, which limits the generalisability of the conclusions.

3. the paper only compares healthy cats to HCM. In humans, this is an easy problem -- the difficulty comes when you also consider other pathologies/phenotypes such as DCM, valve disease etc which probably give a similar cardiac silhouette. I would be interested to know how the algorithm performs on other pathologies.

4. they use the opinion of an experienced reader as ground truth -- this is not very strong. Wouldn't echo/ultrasound give you a much stronger gold standard?

5. the comparison of 5 network architecture does not add much -- particularly since results are identical.

Minor comments

1. while the paper is easy to follow there are several grammatical errors which could do with correction

2. were images augmented for training? Presumably so given the small size of the training data but the parameters need to be stated.

3. Several performance measures are reported (e.g. in table 1), but sensitivity and specificity would be most useful

4. please include more details on ethics.

Reviewer #2: As its name indicates, this paper deals with Deep learning-based diagnosis of feline hypertrophic cardiomyopathy through Comparison of five neural engines

1. I do not have any comments about the deep models, but the outcomes of the simulations are given only quantitatively. It would be nice to have some expert feedbacks for evaluating the results.

2. The authors should also show that the results explained in this article have fair developing stages without making “peeking”. Briefly, peeking is using testing datasets for validation purposes (such as parameter tuning) by making too many iterative submissions (Kuncheva (2014), page 17). In other words, testing sets, which should only be in the previously unseen data, now do not serve testing purposes anymore. Even though it is an indirect usage, peeking is surprisingly an underestimated problem in academic studies which causes overfitting of deep models on a target data. Therefore, theoretically very successful models for specific data may not be useful for real-world problems.

Kavur, A. Emre, et al. "CHAOS challenge-combined (CT-MR) healthy abdominal organ segmentation." Medical Image Analysis 69 (2021): 101950.

The authors should at least discuss the potential effects of peeking on their results.

They should also consider, (maybe apply if possible) and discuss strategies that are being used to avoid peeking such as:

Selver. et al "Basic Ensembles of Vanilla-Style Deep Learning Models Improve Liver Segmentation From CT Images." arXiv preprint arXiv:2001.09647 (2020).

Conze, Pierre-Henri, et al. "Abdominal multi-organ segmentation with cascaded convolutional and adversarial deep networks." Artificial Intelligence in Medicine 117 (2021): 102109.

3. Since the authors used many models to compare the results, they can also test the ensemble strategies, which are shown to outperform single model results at medical image analysis applications such as :

Menze, Bjoern H., et al. "The multimodal brain tumor image segmentation benchmark (BRATS)." IEEE transactions on medical imaging 34.10 (2014): 1993-2024.

Kavur, A. Emre, et al. "Comparison of semi-automatic and deep learning-based automatic methods for liver segmentation in living liver transplant donors." Diagnostic and Interventional Radiology 26.1 (2020): 11.

4. The diversity and omplementarity of the utilized models shoudl also be analyzed through statistis and Kappa. An eemplary analysis an be found at

Toprak, Tugce, et al. "Conditional weighted ensemble of transferred models for camera based onboard pedestrian detection in railway driver support systems." IEEE Transactions on Vehicular Technology 69.5 (2020): 5041-5054.

6. PLOS authors have the option to publish the peer review history of their article (what does this mean?). If published, this will include your full peer review and any attached files.

Reviewer #1: **Yes: **Rhodri H Davies

Reviewer #2: No

---

## [Author Response · Author response to Decision Letter 0]

7 Oct 2022

[2022.09.16.]

Dear Reviewers: 

We thank you and the reviewers for your thoughtful suggestions and insights. The manuscript has benefited from these insightful suggestions. I look forward to working with you and the reviewers to move this manuscript closer to publication in PLOS ONE.

The manuscript has been rechecked and the necessary changes have been made in accordance with the reviewers’ suggestions. The responses to all comments have been prepared and attached herewith/given below. 

Thank you for your consideration. I look forward to hearing from you.

Sincerely,

Hwa-Young Son

College of Veterinary Medicine, 

Chungnam National University, Daejeon, 

Republic of Korea

E-mail: hyson@cnu.ac.kr

Reviewer #1: Summary: Rho et al attempt to use deep learning networks to diagnose hypertrophic cardiomyopathy from radiographs of cats. Networks were trained on 231 radiographs (143 HCM vs 88 normal) and tested on a hold-out dataset of 22 radiographs (11 HCM vs 11 normals). Five different off-the-shelf network types were compared and all five gave identical results: 95% diagnostic accuracy (compared to 75% without AI).

Review:

I disclose that I have no expertise in vetenary/feline medicine but there does seem to be significant utility in such an algorithm. The manuscript is easy to follow and understand.

There are however some limitations:

1. the numbers of both training and test data (only 11 diseased subjects) is small. Probably too small to judge the algorithm's effectiveness on.

- Thank you for your advice. We understand your concerns regarding the small test datasets. Because of the scarcity of legally available radiographic images of HCM-affected cats, we had no choice but to focus on training the model. To enhance the credibility of our model, additional 10 radiographic images of normal and HCM-affected cats from another animal hospital are evaluated.

2. the validation set seems to be from the same institution and not independent, which limits the generalisability of the conclusions.

- Thank you for the critical advice on the generalizability of the image. The image is obtained not only from the teaching hospital but also from three other local hospitals. We did not comment on the data's origin for the ethical and private concerns. Here, we included the tables describing the origin of the images (although the specific name of the institution is hidden) in supplementary table 1.

3. the paper only compares healthy cats to HCM. In humans, this is an easy problem -- the difficulty comes when you also consider other pathologies/phenotypes such as DCM, valve disease etc which probably give a similar cardiac silhouette. I would be interested to know how the algorithm performs on other pathologies.

- We appreciate your advice. The prevalence of cardiac disease in cat is quite different from any other animals or human. The most frequent, predominant cardiac disease is HCM and other cardiac disease is very rare in cat. In 306 primary cardiac disorders from 1998 to 2005, 252 cases were cardiomyopathy (82%) and 48 cases (16%) were congenital heart disease. We unfortunately failed to obtain radiographic images of other cardiac diseases in cats. Therefore, diagnosing whether it is DCM from radiographic image may be most important for cats. 

S.C. Riesen, A. Kovacevic, et al. Prevalence of heart disease in symptomatic cats: an overview from 1998 to 2005

Richard W. Nelson, C. Guillermo Couto, Small animal internal medicine 5th edition.

4. they use the opinion of an experienced reader as ground truth -- this is not very strong. Wouldn't echo/ultrasound give you a much stronger gold standard?

- We thank you for the interesting question. As mentioned, the gold standard for diagnosing heart disease is an ultrasound. 

Because only specialized veterinarians can perform echocardiography and a US machine is required, most veterinarians rely on X-ray and auscultation. The animal is referred to a specialized animal hospital if cardiac disease is suspected in auscultation or radiography. At this point, the deep learning algorithm we developed can suggest and screen whether the animal may be affected by HCM at the x-ray level.

5. the comparison of 5 network architecture does not add much -- particularly since results are identical.

- Thank you for the advice. As the accuracy of the architecture is high predominantly, comparing the five models seems futile. However, in the detailed observation, you will see that misdiagnosed images are different except Resnets. The data shows that each model has different flaws that drive the misdiagnosis. 

Based on your opinion that the results are identical, we ensembled the trained architecture. The ensemble model works when the prediction made by each model considers five prediction values and makes the final decision. As a result, the ensemble model achieved 100% accuracy from the old test sample.

Minor comments

1. while the paper is easy to follow there are several grammatical errors which could do with correction

- Thank you for the detailed advice. We requested a re-assessment of the English and modified it accordingly. 

2. were images augmented for training? Presumably so given the small size of the training data but the parameters need to be stated.

- Thank you for the specific advice. We strengthened the detailed description of data augmentation. 

3. Several performance measures are reported (e.g. in table 1), but sensitivity and specificity would be most useful

- Thank you for the thorough suggestion. We added sensitivity and specificity to the table1.

4. please include more details on ethics.

- We appreciate your advice. We elaborated on the details of the ethics.

Reviewer #2: As its name indicates, this paper deals with Deep learning-based diagnosis of feline hypertrophic cardiomyopathy through Comparison of five neural engines

1. I do not have any comments about the deep models, but the outcomes of the simulations are given only quantitatively. It would be nice to have some expert feedbacks for evaluating the results.

- Thank you for the advice. We invited an expert as an author as participating the revision in the computer vision field. 

2. The authors should also show that the results explained in this article have fair developing stages without making “peeking”. Briefly, peeking is using testing datasets for validation purposes (such as parameter tuning) by making too many iterative submissions (Kuncheva (2014), page 17). In other words, testing sets, which should only be in the previously unseen data, now do not serve testing purposes anymore. Even though it is an indirect usage, peeking is surprisingly an underestimated problem in academic studies which causes overfitting of deep models on a target data. Therefore, theoretically very successful models for specific data may not be useful for real-world problems.

Kavur, A. Emre, et al. "CHAOS challenge-combined (CT-MR) healthy abdominal organ segmentation." Medical Image Analysis 69 (2021): 101950.

The authors should at least discuss the potential effects of peeking on their results.

They should also consider, (maybe apply if possible) and discuss strategies that are being used to avoid peeking such as:

Selver. et al. "Basic Ensembles of Vanilla-Style Deep Learning Models Improve Liver Segmentation From CT Images." arXiv preprint arXiv:2001.09647 (2020).

Conze, Pierre-Henri, et al. "Abdominal multi-organ segmentation with cascaded convolutional and adversarial deep networks." Artificial Intelligence in Medicine 117 (2021): 102109.

- Thank you for the crucial advice. As mentioned, peeking is a crucial obstacle when we develop the model. To avoid peeking, we divided the validation sample from the training sample during the training session and tested it on untrained samples. In addition, we obtained additional radiographic samples from another vet hospital and tested them. Although the sample accuracy obtained was not very precise, we achieved over 78% accuracy from additional samples.

We added these findings to the results section.

3. Since the authors used many models to compare the results, they can also test the ensemble strategies, which are shown to outperform single model results at medical image analysis applications such as :

- We appreciate your important advice. As you mentioned, the ensemble strategy can outperform a single model when every single neural model shows high variance in its prediction. We applied a voting ensemble to five models and achieved 100% accuracy from the test sample. The additional sample showed only 80% accuracy, although the single model (Xception) achieved 97% accuracy.

This finding has been added to the results section.

Menze, Bjoern H., et al. "The multimodal brain tumor image segmentation benchmark (BRATS)." IEEE transactions on medical imaging 34.10 (2014): 1993-2024.

Kavur, A. Emre, et al. "Comparison of semi-automatic and deep learning-based automatic methods for liver segmentation in living liver transplant donors." Diagnostic and Interventional Radiology 26.1 (2020): 11.

4. The diversity and omplementarity of the utilized models shoudl also be analyzed through statistis and Kappa. An eemplary analysis an be found at

Toprak, Tugce, et al. "Conditional weighted ensemble of transferred models for camera based onboard pedestrian detection in railway driver support systems." IEEE Transactions on Vehicular Technology 69.5 (2020): 5041-5054.

- Thank you for the critical advice for our research. In our opinion, the kappa analysis may be helpful in cases wherein there is no exact answer to the problem (ex, unsupervised training). However, the analysis is not applicable in this study because vet specialists confirm all the data. We already compared the results of each model and provided the accuracy of each model.

---

## [Decision Letter · Decision Letter 1]

8 Nov 2022

PONE-D-22-14859R1Deep learning-based diagnosis of feline hypertrophic cardiomyopathy: Comparison of five deep neural network modelsPLOS ONE

Dear Dr. Son

Thank you for submitting your manuscript to PLOS ONE. After careful consideration, we feel that it has merit but does not fully meet PLOS ONE’s publication criteria as it currently stands. Therefore, we invite you to submit a revised version of the manuscript that addresses the points raised during the review process.

ACADEMIC EDITOR:Revise==============================

We look forward to receiving your revised manuscript.

Kind regards,

Jude Hemanth

Academic Editor

PLOS ONE

Journal Requirements:

Reviewers' comments:

Reviewer's Responses to Questions

**Comments to the Author**

1. If the authors have adequately addressed your comments raised in a previous round of review and you feel that this manuscript is now acceptable for publication, you may indicate that here to bypass the “Comments to the Author” section, enter your conflict of interest statement in the “Confidential to Editor” section, and submit your "Accept" recommendation.

Reviewer #1: All comments have been addressed

2. Is the manuscript technically sound, and do the data support the conclusions?

Reviewer #1: Yes

3. Has the statistical analysis been performed appropriately and rigorously? 

Reviewer #1: Yes

4. Have the authors made all data underlying the findings in their manuscript fully available?

Reviewer #1: Yes

5. Is the manuscript presented in an intelligible fashion and written in standard English?

Reviewer #1: Yes

6. Review Comments to the Author

Reviewer #1: Thank you to the authors for taking our comments on board and adapting the manuscript -- a lot of effort has gone into this the manuscript is improved as a result.

In summary, the test data has expanded, but still on the small side (though I appreciate the scarcity of data) and the ground truth is slightly suboptimal (but consistent with what is used in clinical practice). The authors have answered all my comments and I have responded to these, with my responses denoted by '>>'.

1.. the numbers of both training and test data (only 11 diseased subjects) is small.

Probably too small to judge the algorithm's effectiveness on.

- Thank you for your advice. We understand your concerns regarding the small test

datasets. Because of the scarcity of legally available radiographic images of HCMaffected cats, we had no choice but to focus on training the model. To enhance the

credibility of our model, additional 10 radiographic images of normal and HCM-affected

cats from another animal hospital are evaluated.

>> thank you. It would be worth emphasising in the main text that you now have more independent test data (the text still reads 11 HCM and 11 normals).

2. the validation set seems to be from the same institution and not independent, which

limits the generalisability of the conclusions.

- Thank you for the critical advice on the generalizability of the image. The image is

obtained not only from the teaching hospital but also from three other local hospitals.

We did not comment on the data's origin for the ethical and private concerns. Here, we

included the tables describing the origin of the images (although the specific name of

the institution is hidden) in supplementary table 1.

>>thanks. it would be well worth emphasising this in the main text.

3. the paper only compares healthy cats to HCM. In humans, this is an easy problem --

the difficulty comes when you also consider other pathologies/phenotypes such as

DCM, valve disease etc which probably give a similar cardiac silhouette. I would be

interested to know how the algorithm performs on other pathologies.

- We appreciate your advice. The prevalence of cardiac disease in cat is quite different

from any other animals or human. The most frequent, predominant cardiac disease is

HCM and other cardiac disease is very rare in cat. In 306 primary cardiac disorders

from 1998 to 2005, 252 cases were cardiomyopathy (82%) and 48 cases (16%) were

congenital heart disease. We unfortunately failed to obtain radiographic images of

other cardiac diseases in cats. Therefore, diagnosing whether it is DCM from

radiographic image may be most important for cats.

S.C. Riesen, A. Kovacevic, et al. Prevalence of heart disease in symptomatic cats: an

overview from 1998 to 2005

Richard W. Nelson, C. Guillermo Couto, Small animal internal medicine 5th edition.

>>thanks for clarifying -- this makes sense. Could this information (and the references) be added to the main body of the paper?

4. they use the opinion of an experienced reader as ground truth -- this is not very

strong. Wouldn't echo/ultrasound give you a much stronger gold standard?

- We thank you for the interesting question. As mentioned, the gold standard for

diagnosing heart disease is an ultrasound.

Because only specialized veterinarians can perform echocardiography and a US

machine is required, most veterinarians rely on X-ray and auscultation. The animal is

referred to a specialized animal hospital if cardiac disease is suspected in auscultation

or radiography. At this point, the deep learning algorithm we developed can suggest

and screen whether the animal may be affected by HCM at the x-ray level.

>>OK, thanks. Like me, many PLOS readers will also have little expertise in feline medicine and the availability of data so it would be worth emphasising this in the text. It would also be worth adding a discussion about the limitations (subjectivity, error and variability of experts etc) in the discussion.

5. the comparison of 5 network architecture does not add much -- particularly since

results are identical.

- Thank you for the advice. As the accuracy of the architecture is high predominantly,

comparing the five models seems futile. However, in the detailed observation, you will

see that misdiagnosed images are different except Resnets. The data shows that each

model has different flaws that drive the misdiagnosis.

Based on your opinion that the results are identical, we ensembled the trained

architecture. The ensemble model works when the prediction made by each model

considers five prediction values and makes the final decision. As a result, the

ensemble model achieved 100% accuracy from the old test sample.

>>Thanks. You might want to reconsider the title etc. The main contribution to the paper is a novel algorithm for diagnosing feline HCM from X-rays. Adding the subtitle about 5 neural nets downplays your contribution a little...

Thank you for addressing the other comments -- they have been done satisfactorily.

7. PLOS authors have the option to publish the peer review history of their article (what does this mean?). If published, this will include your full peer review and any attached files.

Reviewer #1: No

---

## [Author Response · Author response to Decision Letter 1]

8 Dec 2022

[2022.12.09.]

Dear Editor

PLOS ONE

Dear Editor: 

We wish to re-submit the manuscript titled ““Deep learning-based diagnosis of feline hypertrophic cardiomyopathy: Comparison of five neural engines.” The manuscript ID is D-22-14859.

We thank you and the reviewer for your considerate suggestions and advice. The manuscript has benefited from these insightful suggestions. I look forward to working with you and the reviewers to move this manuscript closer to publication in PLOS ONE.

The manuscript has been rechecked and the necessary changes have been made in accordance with the reviewers’ suggestions. The responses to all comments have been prepared and attached herewith/given below. 

Thank you for your consideration. I look forward to hearing from you.

Sincerely,

Hwa-Young Son

College of Veterinary Medicine, 

Chungnam National University, Daejeon, 

Republic of Korea

E-mail: hyson@cnu.ac.kr

Reviewer #1: Thank you to the authors for taking our comments on board and adapting the manuscript -- a lot of effort has gone into this the manuscript is improved as a result.

In summary, the test data has expanded, but still on the small side (though I appreciate the scarcity of data) and the ground truth is slightly suboptimal (but consistent with what is used in clinical practice). The authors have answered all my comments and I have responded to these, with my responses denoted by '>>'.

1.. the numbers of both training and test data (only 11 diseased subjects) is small.

Probably too small to judge the algorithm's effectiveness on.

- Thank you for your advice. We understand your concerns regarding the small test

datasets. Because of the scarcity of legally available radiographic images of HCMaffected cats, we had no choice but to focus on training the model. To enhance the

credibility of our model, additional 10 radiographic images of normal and HCM-affected

cats from another animal hospital are evaluated.

>> thank you. It would be worth emphasising in the main text that you now have more independent test data (the text still reads 11 HCM and 11 normals).

- Thank you for the advice. We changed the main text to 42 images used in the test dataset.

2. the validation set seems to be from the same institution and not independent, which

limits the generalisability of the conclusions.

- Thank you for the critical advice on the generalizability of the image. The image is

obtained not only from the teaching hospital but also from three other local hospitals.

We did not comment on the data's origin for the ethical and private concerns. Here, we

included the tables describing the origin of the images (although the specific name of

the institution is hidden) in supplementary table 1.

>>thanks. it would be well worth emphasising this in the main text. 

- thank you. The generalizability of the data is added to the abstract and the materials and methods. To emphasize the five institutions, we moved supplementary table 1 into table 1.

3. the paper only compares healthy cats to HCM. In humans, this is an easy problem --

the difficulty comes when you also consider other pathologies/phenotypes such as

DCM, valve disease etc which probably give a similar cardiac silhouette. I would be

interested to know how the algorithm performs on other pathologies.

- We appreciate your advice. The prevalence of cardiac disease in cat is quite different

from any other animals or human. The most frequent, predominant cardiac disease is

HCM and other cardiac disease is very rare in cat. In 306 primary cardiac disorders

from 1998 to 2005, 252 cases were cardiomyopathy (82%) and 48 cases (16%) were

congenital heart disease. We unfortunately failed to obtain radiographic images of

other cardiac diseases in cats. Therefore, diagnosing whether it is HCM from

radiographic image may be most important for cats.

S.C. Riesen, A. Kovacevic, et al. Prevalence of heart disease in symptomatic cats: an

overview from 1998 to 2005

Richard W. Nelson, C. Guillermo Couto, Small animal internal medicine 5th edition.

>>thanks for clarifying -- this makes sense. Could this information (and the references) be added to the main body of the paper?

- thank you for the advice. this information is added to the introduction.

4. they use the opinion of an experienced reader as ground truth -- this is not very

strong. Wouldn't echo/ultrasound give you a much stronger gold standard?

- We thank you for the interesting question. As mentioned, the gold standard for

diagnosing heart disease is an ultrasound.

Because only specialized veterinarians can perform echocardiography and a US

machine is required, most veterinarians rely on X-ray and auscultation. The animal is

referred to a specialized animal hospital if cardiac disease is suspected in auscultation

or radiography. At this point, the deep learning algorithm we developed can suggest

and screen whether the animal may be affected by HCM at the x-ray level.

>>OK, thanks. Like me, many PLOS readers will also have little expertise in feline medicine and the availability of data so it would be worth emphasising this in the text. It would also be worth adding a discussion about the limitations (subjectivity, error and variability of experts etc) in the discussion.

- thank you for the advice. The feline medicine and the availability of the data is added into the discussion. 

5. the comparison of 5 network architecture does not add much -- particularly since

results are identical.

- Thank you for the advice. As the accuracy of the architecture is high predominantly,

comparing the five models seems futile. However, in the detailed observation, you will

see that misdiagnosed images are different except Resnets. The data shows that each

the model has different flaws that drive the misdiagnosis.

Based on your opinion that the results are identical, we ensembled the trained

architecture. The ensemble model works when the prediction made by each model

considers five prediction values and makes the final decision. As a result, the

ensemble model achieved 100% accuracy from the old test sample.

>>Thanks. You might want to reconsider the title etc. The main contribution to the paper is a novel algorithm for diagnosing feline HCM from X-rays. Adding the subtitle about 5 neural nets downplays your contribution a little...

- Thank you for the advice. We changed the title so that the novel algorithm and the ensemble method are focused.

---

## [Decision Letter · Decision Letter 2]

2 Jan 2023

Deep learning-based diagnosis of feline hypertrophic cardiomyopathy

PONE-D-22-14859R2

Dear Dr. Son

We’re pleased to inform you that your manuscript has been judged scientifically suitable for publication and will be formally accepted for publication once it meets all outstanding technical requirements.

Kind regards,

Jude Hemanth

Academic Editor

PLOS ONE

Additional Editor Comments (optional):

Reviewers' comments:

Reviewer's Responses to Questions

**Comments to the Author**

1. If the authors have adequately addressed your comments raised in a previous round of review and you feel that this manuscript is now acceptable for publication, you may indicate that here to bypass the “Comments to the Author” section, enter your conflict of interest statement in the “Confidential to Editor” section, and submit your "Accept" recommendation.

Reviewer #1: All comments have been addressed

2. Is the manuscript technically sound, and do the data support the conclusions?

Reviewer #1: Yes

3. Has the statistical analysis been performed appropriately and rigorously? 

Reviewer #1: Yes

4. Have the authors made all data underlying the findings in their manuscript fully available?

Reviewer #1: Yes

5. Is the manuscript presented in an intelligible fashion and written in standard English?

Reviewer #1: Yes

6. Review Comments to the Author

Reviewer #1: Thank you answering my queries. All my comments have been addressed.

Congratulations on a nice piece of work

7. PLOS authors have the option to publish the peer review history of their article (what does this mean?). If published, this will include your full peer review and any attached files.

Reviewer #1: No

---

## [Editor Report · Acceptance letter]

24 Jan 2023

PONE-D-22-14859R2 

Deep learning-based diagnosis of feline hypertrophic cardiomyopathy 

Dear Dr. Son:

I'm pleased to inform you that your manuscript has been deemed suitable for publication in PLOS ONE. Congratulations! Your manuscript is now with our production department. 

Kind regards, 

on behalf of

Dr. Jude Hemanth 

Academic Editor

PLOS ONE